# The In Vitro and In Vivo Synergistic Antimicrobial Activity Assessment of Vacuum Microwave Assisted Aqueous Extracts from Pomegranate and Avocado Fruit Peels and Avocado Seeds Based on a Mixtures Design Model

**DOI:** 10.3390/plants10091757

**Published:** 2021-08-24

**Authors:** Prodromos Skenderidis, Stefanos Leontopoulos, Konstantinos Petrotos, Chrysanthi Mitsagga, Ioannis Giavasis

**Affiliations:** 1Laboratory of Food and Biosystems Engineering, Department of Agrotechnology, University of Thessaly, 41110 Larissa, Greece; sleontopoulos@uth.gr (S.L.); petrotos@uth.gr (K.P.); 2Laboratory of Food Microbiology, Department of Food Technology, University of Thessaly, End of N. Temponera Street, 43100 Karditsa, Greece; cmitsanga@uth.gr (C.M.); igiavasis@uth.gr (I.G.)

**Keywords:** avocado, pomegranate, antimicrobial activity, microwave assisted extraction, lyophilisation, encapsulation

## Abstract

The present study aimed to assess the antimicrobial properties of encapsulated lyophilized powdered extracts of pomegranate peels (PP), avocado peels (AP) and avocado seeds (AS) in vitro and in vivo. Minimum Inhibitory Concentration (MIC) and Minimal Bactericidal Concentration (MBC) methods, optical density measurement, and well diffusion assay were used to determine antimicrobial activity against food borne bacteria (Gram− *Escherichia coli, Salmonella typhimurium, Campylobacter jejuni, Pseudomonas putida*), (Gram+ *Staphylococcus aureus, Listeria monocytogenes, Clostridium perfringens, Lactobacillus plantarum*), and fungi (*Penicillium expansum and Aspergillus niger*) based on a mixture design model. Additionally, the most effective powder was studied in vivo in yogurt, cream cheese, and minced meat burger. The samples that contained high polyphenol content also exhibited higher antioxidant, antimicrobial, and antifungal activity. From the results of the well diffusion, the MIC/MBC, and the cell optical density assays, the antimicrobial activity of the extracts was found to be correlated to the total phenolic content (TPC) of the samples and the type of the microorganism. The pomegranate peels extract presented the higher TPC and antioxidant activity and constitute the highest percentage in the most active antimicrobial mixture. The powders that were tested in vitro showed microbial type-dependent effects in each food model. The results presented here can be further studied in the large-scale industrial production of natural food preservatives.

## 1. Introduction

There is currently a growing interest in developing alternatives to synthetic antimicrobial agents widely used in the food and pharmaceutical industries. This is due to consumer interest in the safety of materials containing synthetic chemicals and the health dangers associated with them [1]. Many fruit by-products like peels and seeds that contain a high concentration of bioactive organic compounds are generally disposed of as waste, resulting in an unsolvable environmental problem [2].

Two important plant species considered to be functional with many bioactive substances are *Persea americana* (avocado) and *Punica granatum* (pomegranate) [3,4]. The avocado plant (*Persea americana* Mill.) is grown worldwide in the tropics, and Mexico is the country that leads the production of avocado fruits, which are berries with a single large seed [3]. Avocados often referred to as a “super-food” due to their known high nutrient content and health benefits, which are essentially due to the source of fat-soluble nutrients or phytochemicals [5,6,7]. The avocado pulp contains higher amounts of insoluble and soluble fiber, protein and sugars, pigments, tannins, polyphenols, phytoestrogens percitol [8,9,10,11,12], lipids, and oil [10,13] compared with the contents found in other fruits. During the processing of avocado fruit, peel and seed by-products, (comprising the 30% of the total fruit weight) are generally disposed of as waste [14]. Based on the high concentrations of these bioactive substances, peels and seeds have been used in cosmetics, food, and the medical industry [15]. It has been reported that the seeds have some anti-nutritional characteristics such as hydrocyanic acid, cyanogenic glycosides, condensed polyphenols and some tannins, which could act adversely on their possible use. Seeds are also reported to contain a natural, oil-soluble fungicide called persin (belonging to biologically active aliphatic acetogenin) which is harmless to humans but can be toxic to birds and some animals [16,17].

Pomegranate (*Punica granatum*) is a fruit with a worldwide distribution [18]. The fruit can be consumed either directly or after processing where its juice is extracted. Pomegranate juice waste is the result of the productive process of squeezing the fruit to absorb its juice. This industrial activity has social and economic importance for the agricultural areas where arboricultural species are grown, but also ecological due to large quantities of waste produced each growing season from the processing of the fruits. Such wastes are mainly the ground-crushed solid components of the fruit (peel and seeds), representing about 50% of the total weight [19]. As a result, pomegranate peel may be used in food industry applications due to its high concentration in bioactive compounds like punicalagin, which have exhibited stronger biological activities than pomegranate juice [20,21,22]. Punicalagin is a water-soluble polyphenol (ellagitannin) that includes a gallic acid portion linked to a glucose molecule. It has presented antioxidant, antifungal, and antibacterial properties [23].

The extraction method, as well as the solvent and extraction conditions (e.g., temperature, time) used, are important factors considering the extract’s final composition in bioactive compounds and their antimicrobial properties [24].

Vacuum Microwave-assisted aqueous extraction is characterized by shorter extraction time, smaller size of the required equipment, higher efficiency, reduced energy consumption, lower amount of waste and shorter exposure of bioactive components to thermal destruction [25,26]. The use of water as the extraction solvent decreases the concentration of persin in the received extracts since persin is a lipophilic compound [17].

The process used to dry the extracts into powder form is an important factor that determines their bioactivity. Freeze-drying (lyophilisation), which is performed in this experiment, is one of the most effective methods for shielding thermo sensitive and dysfunctional molecules such as fruit polyphenols, maximizing their bioactivity [27,28]. 

However, in order to achieve more sufficient bioactivity of natural antimicrobials, many auxiliary factors must be taken into account. One of these factors is the stability and the slow release of the antimicrobial agent in the application medium. The above object can be achieved using micro-encapsulation of the antimicrobial agent [29]. Encapsulation is a technique by which active solid, liquid, or gas compounds are introduced into a matrix of the polymeric wall system to protect them from environmental conditions (light, oxygen, temperature, and water), avoiding oxidation, increasing shelf life, preserving interactions with other food components, or controlling their release for a specific place and/or time [30].

It has been reported that the mixture of the compounds contained in a herbal product show a significantly higher synergistic effect than an equivalent dose of a single, isolated active ingredient or a single bioactive plant compound [31,32]. The bioactive effects of such mixtures could be a result of the total sum of different classes of compounds with diverse mechanisms of action [33].

Despite the studies on the antimicrobial activity of avocado [34,35] and pomegranate [36,37] fruits by-products, there is no study on the evaluation of the synergistic antimicrobial activity of these by-product mixtures. Hence, this study aimed to evaluate the potential synergist antimicrobial activity of aqueous vacuum microwave assisted extracts from avocado peels, seeds, and pomegranate peels in liquid and powder form in vitro against food-born bacteria, yeasts, and fungus based on a mixture design experiment. Additionally, the effective samples were tested as natural preservatives in food models of minced meat, cheese salad dressing, and yogurt. The novelty of this study is the focus of the sustainable valorization of pomegranate and avocado wastes studying their synergistic effects to produce an antimicrobial natural preservative against important bacterial and fungal species involved in food spoilage. The in vitro and in vivo antimicrobial evaluation of the waste origin from Greek avocado and pomegranate process companies showed potential advantages for their use natural antimicrobial agents in the food industry.

## 2. Results

From the analysis of the TPC and the antioxidant activity based on the estimation of the IC_50_ of the DPPH radical scavenging, it is clear that PP have a significantly higher concentration of polyphenols and antioxidant ability than avocado wastes. Comparing the two avocado waste results, AS shows a lower TCP and higher IC_50_ of DPPH as can be seen in Table 1. Additionally, a significant Pearson correlation of samples’ TPC with their antioxidant activity (based on the IC_50_ of DPPH radical scavenging) was estimated at −0.92 (the negative correlation explained from that the lower DPPH IC_50_ the higher antioxidant activity) with a *p*-value < 0.001 (results not shown).

The in vitro evaluation of the antimicrobial effects against bacteria, yeast, and fungi was performed. From the well diffusion assay and inhibition zones, results were observed only for fungus *A. niger, P. expansum,* and the *C. jejuni*. Fungus well diffusion assay results and inhibition zones of the *L. monocytogenes* and *C. perfringens* are presented in Table 2 and Table 3. Based on these results, the growth of fungus *A. niger* was affected by the addition of sample No6 (100% PP), leading to the lower size of mycelium at the 3 d (4.37 mm), 5 d (10.32 mm) and 7 d (11.82 mm) measurements. The second most effective sample was the No7 (50% PP + 50% AP) which exhibited significantly lower growth even at the 5 d (6.35 mm) and higher at the 3 d and 7 d (17.42 mm) compared to the sample No6. Regarding the fungus *P. expansum* No6 also showed similar inhibited fungus growth at 3 d (4.59 mm) while sample No7 showed higher inhibition in the growth of mycelium (6.35 mm and 9.34 mm) at 5 d and 7 d respectively.

The effects on the growth of the gram-positive bacteria based on the inhibition zones method showed that *L. monocytogenes* was affected only by the samples No6 (2.95 mm) and No9 (2.75 mm) while *C. perfringens* strain No7 showed the largest inhibition zone (4.6 mm) followed by sample No 9 (3.93 mm). This is in line with the results reported in previous studies that the gram-positive bacteria were more sensitive in comparison to the Gram-negative bacteria [4,12].

Based on the mixture design analysis of these results, the optimum mixtures to achieve the best antimicrobial activity for *A. niger, P. expansum,* and the *C. jejuni**,* individually for each microorganism, were 100% PP, 49.5% PP + 51% AP and 65% PP + 35% AP.

The results from MIC/MBC tests of all samples are summarized in Table 4. Furthermore, a Pearson correlation of each sample’s TCP with their MIC was performed and presented in Table 5.

Among the samples tested for MIC, we observed that sample No6 (100% PP) and No10 (50% PP + 50% AS) inhibit the growth of all microorganisms (except the *P. expansum* and *P. putida*) in the higher examined concentration having an MIC from 25 to 50 mg/mL based on the tested bacteria (Table 5). More specifically, the MIC of *S. typhimurium*, *L. monocytogenes*, *C. perfringens* and *C. utilis* was the maximum examined concentration (50 mg/mL) while *E. coli, S. aureus, C. jejuni* and *L. plantarum* showed half the value of MIC at 25 mg/mL.

The only differentiation between the MIC of the two samples presented for *A. niger* was that sample No6 exhibited better antimicrobial activity, achieving half the MIC (25 mg/mL) in comparison to sample No10 (50 mg/mL) while the MIC of sample No6 for *P. expansum* was 50 mg/mL. The results showed that the MIC required for the *C. jejuni* and the *L. plantarum* was the same.

Furthermore, the MIC of samples No9 and No10 were similar for *E.*
*coli, C. jejuni, S. aureus, L. monocytogenes,* and *L. plantarum*. However, different results were observed for *A. niger* where sample No9 showed lower MIC than No10. Differentiations in MIC of the two samples were also presented for *P. expansum, C. utilis, C. perfringens* and *S. typhimurium*.

The mixture design analysis performed by Design Expert statistical tool and the optimization showed that a mixture containing 52% PP 1.5% AP and 46.5% AS is needed in order to achieve the optimal antimicrobial activity based on the MIC analysis for all microorganisms (Figure 1).

According to the turbidity assay results, sample No6 was shown to be the most effective at leading to a reduction of the absorbance at 620 nm as an effect of the addition of each sample at concentrations of 10 mg/mL, compared to the control (0 mg/mL) without antimicrobial extracts (Figure 2a–i).

With regard to the effects of the samples No6, No7, No9, and No10 from the in vivo studies in the minced meat model, the results show that sample No7 achieved a higher reduction of TMC on the 4th and 6th day. In addition, the same sample achieved a higher reduction effect in *E. coli* while sample No6 achieved a higher reduction effect against *C. jejuni* on the 4th day and sample No10 on the 6th day (Figure 3).

In the yogurt food model, all the tested samples presented better reduction effects against yeasts and fungus compared to the effects presented by using sorbic acid which was used as control. It was observed that sample No10 showed the best effects for yeasts and fungus on the 7th and 14th day while samples No9 and No10 showed similar results to sorbic acid antimicrobial results on the 7th day against *P. expansum*. On the 14th and 21st days, samples No6 and No10 achieved the best reduction effects against *P. expansum*.

In the cheese salad model, sample No6 presented the best antimicrobial effect against yeasts and fungus compared to the control. In addition, sample No9 showed the best reduction effect against *R. mucaliganosa* on the 7th day while sample No6 again achieved the best effects on the 14th and 21st days.

## 3. Discussion

Plant extracts’ antimicrobial activity may be attributed to a range of phytochemical substances, including essential oils, terpenoids, alkaloids, lectins, polypeptides, polyphenolics and phenolic compounds (simple phenols, phenolic acids, quinones, flavones, flavonols and flavonoids, tannins and coumarins).

Polyphenols are plant-derived compounds that resulted from plant metabolism. They follow the phenylpropanoid and acetate pathways, especially after exposure to stressful climatic conditions and pathogenic microorganisms, which play an important role in plant resistance [38]. The definition of polyphenol includes a wide range of complex substances of plant origin bearing at least one benzoic ring as well as at least one hydroxyl group attached to a carbon of its cyclic moiety. The usual form with which they occur in nature is glycoside, and their solubility varies. Τhe antimicrobial properties of certain classes of polyphenols have been proposed as a means to develop new natural food preservatives due to the increasing consumer pressure for clean food labels with no addition or the minimal use of harmful synthetic additives [37,39]. However, this activity will always depend on the bacteria species. The polyphenolic profile of PP consists mainly of tannins such as gallotannins and ellagitannins, including the predominant hydrolysable tannin known as punicalagin [23]. The antimicrobial activity of avocado fruit by-products AP and AS is mainly constituted by the flavonoids of procyanidin A and B [14,40] catechins, quercetin, glycerides, triamcinolone acetaminophen, saponins, steroids, caffeoalkinic acid and coumaric acid. This constitutes the main complex mixture of polyphenolic compounds found in avocado residues [15,26,41].

The antimicrobial mechanisms of A-type proanthocyanin on bacterial growth inhibition are based on their effects on the cytoplasmic membrane destabilization, the cell membrane permeabilization, the inhibition of extracellular microbial enzymes, the direct actions on microbial metabolism, or the deprivation of microbial growth substrates, particularly essential minerals [39,42].

Furthermore, similar research completed in berries found that ellagitannins exhibited antimicrobial activity to different extents in the growth of selected Gram-negative intestinal bacteria (strains of Salmonella, Staphylococcus, Helicobacter, *E. coli*, Clostridium, Campylobacter, and Bacillus), but they are not active against Gram-positive beneficial probiotic lactic acid bacteria [43]. Gallotannins presented antibacterial activity against food-borne bacteria and Gram positive bacteria were generally more susceptible than Gram-negative [44]. The activity of gallotannins is attributable to their strong affinity for iron and it is also related to the inactivation of membrane-bound protein.

More specifically, it has been reported that PP increased the preservation time by 2–3 weeks during chilled storage of chicken products [45], by the inhibition of *E. coli, B. cereus, L. monocytogenes, S. aureus, and Y. enterocolitica* [4,46,47,48]. Additionally, many studies have shown the strong antimicrobial activity of flavonols against Gram-positive bacteria, such as *S. aureus* and *L. acidophilus* and Gram-negative bacteria, such as *P. oralis, P. melaninogenica, P. gingivalis, and F. nucleatum,* probably due to different mechanisms of action, mainly on the bacterial cells [39,49].

The results of this study validated the antimicrobial activity of PP as they form the basis of most powerful antimicrobial mixtures. Additionally, the results have shown the synergistic effects of PP polyphenols compounds with antimicrobial active compounds from by-products of avocados processes. In an overall view, it is also validated that the gram-positive bacterium was more sensible to avocado extracts than gram-negative [35].

## 4. Materials and Methods

### 4.1. Plant Materials

Pomegranate fruits “Wonderful” variety was collected from a plantation located in the region of Thessaly, Greece. The pomegranate peel (PP) was collected as a by-product from the process at Rodones S.A. company. The frozen PP mechanically were milled using a commercial mill and prepared in 2 Kg and kept at −28 °C until the extraction. Subsequently, avocado fruit, variety “Pinkerton” collected in 2019 from the region of Crete in Greece, was used in this study. The avocado fruits were peeled and the by-products (peel and seeds) were separated from the fruit. The by-products were then mechanically shivered using a sphere mill to produce the suitable raw material for the extraction. The extraction conditions applied for avocado and pomegranate by-products were described in our previous studies [25,26].

### 4.2. Mixture Design and Antimicrobial Extracts/Samples Composition and Coding

The D-optional method in the mixture design, provided by the software Design-expert (ver. 7.0.0, Stat-Ease Int. Co., Minneapolis, MN, USA), was used to optimize the formulation of the above-received extract powders. Generally, the mixture design is used to study the relationships between the proportion of different variables and responses. Ever since Scheffe devised a single-lattice and single-core design in 1958 [50], the mixture design has developed a variety of methods [51,52]. In our study, a simplex-centroid design with three axial mixture points was used to investigate the effects of the different proportion and the produced lyophilized powders of pomegranate peel, avocado peel and seed extracts on the tested microorganisms (Figure 4). The liquid extracts were divided into three groups of pomegranate peel (PP) A, avocado peel (AP) B, and avocado seeds (AS) C. The selection of the proportion of each group was made by software considering the selectivity triangle, since extracts from different fruit parts with a different concentration in bioactive substances in the triangle may have different antimicrobial ability [53,54].

This method can not only establish the surface model of continuous variables, estimating every element in the mixture and their interactions but can also optimize the component elements according to the target to determine the best ratio of each powder. This method is now widely used for the formulation in the pharmaceutical and food industries and has achieved good results [55,56]. In total, 10 liquid by-product extract mixtures based on the percentage proportion of each are presented in Table 6.

### 4.3. Encapsulation and Lyophilisation of the Extracted Pomegranate and Avocado Peel and Seed Material Mixtures

For the encapsulation, 5% *w/v* of silicon dioxide (SiO_2_) was also added to increase the glass transition temperature (Tg) by optimizing the thermal properties of the system [43].

The lyophilisation process of the 10 mixtures derived from the design mixture was performed in a Zirbus GmbH Sublimator 4 × 5 × 6 freeze dryer and involved three steps in a 24-h cycle process. In the first step (demands approximately two hours), the product was cooled to the temperature of −30 °C. The second step (required approximately 12 h) involved heating the product to a shelf temperature of 0 °C under a vacuum of 0.15 mbar. In the final step, the product was heated to a shelf temperature of 40 °C under a vacuum of 0.15 mbar for approximately 10 h.

The lyophilized samples were sealed in opaque plastic flasks and stored at a constant temperature oven at 30 ± 0.3 °C for up to 15 days to avoid Tg degradation.

### 4.4. Chemicals

Folin-Ciocalteu 2N, anhydrous crystal-formed sodium carbonate (PubChem CID: 10340); gallic acid (PubChem CID: 370); 2,2-diphenyl-1-picrylhydrazyl (DPPH•) (PubChem CID: 74358); methanol (PubChem CID: 887). Chemicals used in this study were supplied from Sigma Aldrich ((St. Louis, MO, USA).

### 4.5. Determination of Total Phenols Content (TPC) of the Extracts

In this work, a previously reported approach was used [57]. In brief, 1.58 mL of distilled water and 100 L of Folin-Ciocalteu reagent (0.2 N) were combined with 20 L of each extract. Following that, 300 L of Na_2_CO_3_ solution (200 g/L) was added, and samples were incubated in the dark for 30 min at 43 °C. An Evolution 201 spectrophotometer from Thermo scientific (Thermo Fisher, Waltham, MA, USA) used for the absorbance measurement at 765 nm and the results expressed as gallic acid equivalent (GAE) based on a calibration curve of the absorbencies of the standard gallic acid in concentrations of 50 to 500 mg/L using for the measurement an EW 220-3NM electronic scale (Kern and Sohn GmbH., Balingen, Germany).

### 4.6. Determination of Antioxidant Capacity of the Samples (DPPH• Method)

Antioxidant capacity was determined based the activity of the extracts to scavenge the active radical of DPPH• (2,2-diphenyl-1-picrylhydrazyl) [58]. Briefly, 50 μL of aqueous dilutions of the samples (mass measurements were taken with an EW 220-3NM electronic scale (Kern and Sohn GmbH., Balingen, Germany)), 900 μL of methanol and 50 μL of DPPH• active radical (freshly made) was added and mixed. The solutions were incubated for 20 min in the dark at 25 °C. Then the incubation the absorbance was measured at 517 nm with the use of an Evolution 201 spectrophotometer from Thermo scientific (Thermo Fisher, Waltham, MA, USA). The solutions of the extracts and the DPPH• methanol solutions were used as blank and control respectively. The trials were performed three times. The percentage of radical-scavenging capacity (RSC) of the tested extracts was calculated using the equation below (Equation (1)):% DPPH•RCS = ((Abs control − Abs sample)/Abs control) × 100(1)
where: Abs control and Abs sample are the absorbance values of the control and the tested sample respectively. To measure the radical-scavenging capability of the extracts, the value of the half-maximum inhibitory concentration (IC_50_) was determined using the graph plotted of the % RSC versus the PPE concentration.

### 4.7. Microorganisms Used for the Determination of Antimicrobial Activity

Pathogenic and spoilage microorganisms used in this study were obtained from different culture collections. These included the pathogenic bacteria *E. coli* ATCC 8739, *S. typhimurium* ATCC 14028, *C. jejuni* ATCC 33291, *S. aureus* ATCC 6538, *L. monocytogenes* ATCC19115, *C. perfringens* ATCC 13124. The fungus *A. niger and P. expansum* were kindly provided by the Greek Benaki Phytopathological Institute (Athens, Greece) culture collection. Foodborne pathogens used in this study, like *E. coli*was cultivated on TBX agar (Oxoid, UK) at 37 °C for 24 h, while *S.*
*aureus* was plated on Baird Parker agar (Oxoid, UK) with the addition of egg yolk tellurite (50 mL/1 L of substrate) (Oxoid, UK) at 37 °C for 48 h. Furthermore, *L. monocytogenes* was counted on Harlequin Listeria Chromogenic Agar (Ottaviani and Agosti) (Neogen, UK), *C. jejuni* was spread on Campylobacter Selective agar (Acumedia-Neogen, USA) with the addition of the supplement CCDA (Oxoid, UK) at 37 °C for 72 h under microaerophilic conditions, *S. typhimurium* was cultivated on XLD (LabChem, Spain) at 37 °C for 24 h and *C. perfringens* was counted on TSC agar (Oxoid, UK) at 37 °C for 48 h under anaerobic conditions. The microorganisms *P. expansum* and, *A. niger* were cultivated on Potato Dextrose agar (Oxoid, UK) at 25 °C for 3–5 days. All the bacteria were also cultivated in liquid cultures using Tryptone Soy Broth (Oxoid, UK) as substrate and incubation as described above for each microorganism. For fungi, Potato Dextrose Broth (Oxoid, UK) was used.

### 4.8. Antimicrobial Assays

#### 4.8.1. Well Diffusion Assay

For the determination of inhibition zones after microbial growth in Petri dishes, the well diffusion assay was used, following the method described by Reller [59]. The presence of only bacteria or yeasts respectively without the addition of any antimicrobial substance was also used as a control. These concentrations and mixtures are described in Table 1. Glass tubes were then prepared with 10% of each powder (1 g of powder diluted in 10 mL of distilled water) and sterilized in an autoclave at 121 °C for 15 min. Then, approximately 20 mL of the appropriate nutrient medium as described in section (Microorganisms used for the determination of antimicrobial activity) was applied on Petri dishes. After the stabilization of the substrates, 0.1 mL of the culture was spread on the Petri dish from each bacterium and yeast that was previously cultivated in Tryptone Soy broth substrate for bacteria and Potato Dextrose broth for yeasts. This was followed by the formation of small wells with the use of a sterile glass Paster pipette in the center of the plate, to which 25 μL of the 10% of the encapsulated powders was added. The plates were then placed in an incubator or left at room temperature according to the needs of each microorganism as described in Section 4.7. Finally, the inhibition zones (expressed in mm) were measured using an electronic calliper. For the *A. niger* and the *P. expansum* fungus before the application of the PDA nutrient into Petri dishes, 1.5 mL of each sample was placed in them. Then 20 mL of PDA nutrient substrate was applied to each plate, stirring simultaneously to incorporate the solution into the nutrient substrate. The plates were then allowed to stand for about 30 min to stabilize the nutrient substrates. Finally, 25 μL of the tested fungus was added into the prepared wells. The number of replicates per treatment was 4. Measurements of the size of mycelium in mm were taken at 3.5 and 7 days of incubation.

#### 4.8.2. MIC/MBC Assay

The above-mentioned pathogenic bacteria were inoculated in test tubes with 9.9 mL liquid growth medium (TSB) supplemented with the antimicrobial samples/extracts to determine the MIC of each sample. More specifically, the test tubes with growth medium contained 0 mg/mL (control), 5, 10, 25 and 50 mg/mL of the above powder samples. These concentrations were chosen based on results of similar studies examining the antimicrobial activity of natural antimicrobials, where a minimum of 10 mg/mL was necessary for expressing antimicrobial activity, even in non-encapsulated samples [60,61]. One milliliter of paraffin was also added to the test tubes for incubation of *C. perfringens* and *C. jejuni* to facilitate anaerobic and microaerophilic incubation, respectively. The test tubes were then sterilized and subsequently inoculated with 0.1 mL of 48 h grown cultures of each of the six pathogenic bacteria, resulting in a 1/100 dilution of the cell suspension and incubated at 37 °C for 48 h to determine MIC, which was defined as the lowest antimicrobial sample concentration that resulted in the absence of growth (absence of cell debris, turbidity or biofilm) of the inoculated microorganism after incubation [62].

MBC was determined after inoculation of 1 mL from the MIC tube (i.e., the test tube with the lowest inhibitory concentration), as well as from all tubes of higher concentration of antimicrobials, into test tubes containing 9 mL of the same growth medium but without any antimicrobial agents. The test tubes were again incubated at 37 °C for 48 h and observed for the presence of microbial growth. MBC was the lowest concentration where no growth was observed.

#### 4.8.3. Optical Density Assay

For the optical density, tubes were prepared with liquid nutrient medium TSB for bacteria and PDB for yeasts in which different concentrations of the encapsulated powders were diluted. The concentrations were 0%, 1%, 2.5% and 5%. The tubes were then sterilized at 121 °C for 15 min. After the sterilization, the tubes were inoculated with 0.1 mL of pathogenic and spoilage bacteria as well as yeasts and incubated under appropriate conditions. Also, tubes containing only substrate (without the addition of any substance) were inoculated with the target microorganisms (control samples) [63]. At the end of the incubation, the samples were diluted in a ratio of 1/10 with deionized water as they had a very dark color which would affect our measurements, and they were spectrophotometrically measured in a Visible-Ultraviolet light spectrophotometer (DR 5000, Hach Lange) at 620 nm. Along with the inoculated tubes we also had tubes that we did not inoculate with the microorganisms so that at the end of our measurements we could remove the color of each powder from our sample to have only the growth of each microorganism.

#### 4.8.4. Preparation of Foods Used in the Food Model

Minced beef meat, ready to eat cheese salad dressing (AMBROSIA, Thessaloniki, Greece) and yogurt were the foods used for the food model. The minced meat and the cheese salad dressing were bought from the local market and the yogurt was produced in the lab based on a Greek-style yogurt recipe with the use of pasteurized cow milk bought from the local market. The nutritional value of all foods is presented in Appendix A. For the in situ antimicrobial activity evaluation of extracts, four samples (No6, No7, No9 and No10 showed the best antimicrobial activity based on the previous methods) were tested in the food models in concentration of 0.5%. This concentration was selected based on experiments on samples’ effects on organoleptic characteristics in yogurt. Concentrations above 0.5% were not desired as changes in color and the taste of yogurt were observed. Furthermore, 0.5 g of each sample (No6, No7, No9 and No10) was diluted in 5 mL deionized water and vortexed for 1 min and the added in 95 g of minced meat and cheese salad dressing tested foods under aseptic conditions. For the positive control, only 5 mL deionized water was added. For the negative control, 90 mg of sorbic acid was used. The same quantities, added in the milk, were used for the yogurt preparation before the pasteurization stage. During the study period, all food samples were persevered at 4 °C in food trays covered with protective film. The trials were performed in triplicate. Samples from each treatment were collected at various intervals (7th 14th and 21st day) and subjected to microbiological analysis.

### 4.9. Statistical Analysis

Standard deviation (SD) was calculated and the averaged values along with the SDs are documented in their respective tables or figures. Statistical differences among the means, as well as interactions between the variables, used in chemical analyses, were detected through one-way ANOVA followed by a Tukey’s test, and statistical significance was set at the *p*
≤ 0.05 level. MiniTab^®^17.1.0 and Design Expert^®^7.0.0 software were used as the tool to perform the design mixture analysis and the above-mentioned tests.

## 5. Conclusions

The by-products from the food processing industry have received significant attention from the scientific community in terms of the circular economy because they are a source of added-value phytochemicals, notably phenolic compounds, which are known to have major antioxidant and antibacterial effects. In this work, the in vitro and in vivo synergistic antimicrobial activity of aqueous extracts of pomegranate fruit peel and avocado fruit peel and seed were extracted with a vacuum microwave assisted extraction method. We showed antimicrobial activity based not only on their polyphenol content but also on the tested microorganism based on the nature of the polyphenol content and their synergy. Among the examined fruit wastes, the extract from pomegranate peel was presented as the base of these mixtures since, in most cases, it constituted nearly half a percentage. More specifically, pomegranate has a significantly higher concentration of polyphenols and higher antioxidant ability compared to avocado waste. Thus, this extract constitutes the basis of the most effective antimicrobial sample. The in vitro tested powders showed microbial type-dependent effects in each food model while the most effective powder studied in vivo was found in yogurt, cream cheese, and minced meat burger. In more detail, all tested samples in the yogurt food model presented better reduction effects against yeasts and fungus compared to the effects presented by using sorbic acid which was used as control. In the cheese salad model, sample No6 presented the best antimicrobial effect against yeasts and fungus compared to the control. The use of plant waste as a source of bioactive compounds is an area of relevance for the development of novel products and this study reveals the potential of avocado and pomegranate waste as a novel source of bioactive compounds for future application in various areas including the agri-food industry. This not only offers an alternative solution to the food industry waste disposal problem with a sustainable approach, but will also help the search for cost-effective natural preservatives at the industrial level. However, future research is needed to examine the link between the antimicrobial effects and the concentration of specific polyphenolic compounds and their variation in these by-products.

## Figures and Tables

**Figure 1 plants-10-01757-f001:**
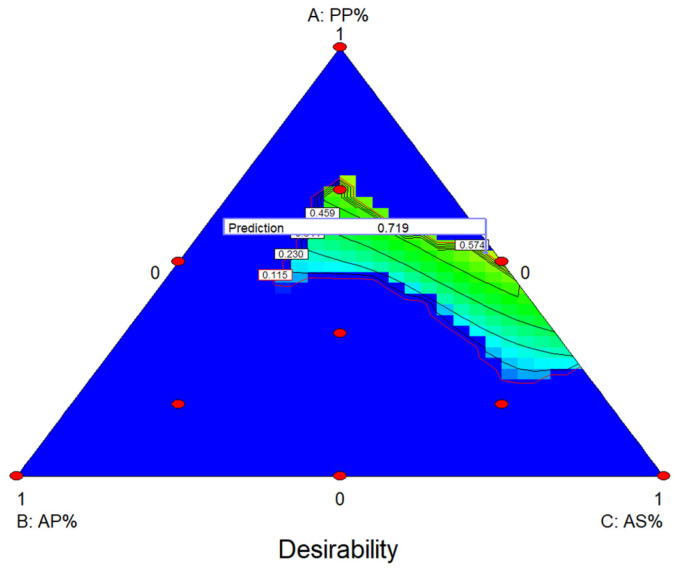
Contour plot of the effect of the samples on MIC based on optimization from mixture design.

**Figure 2 plants-10-01757-f002:**
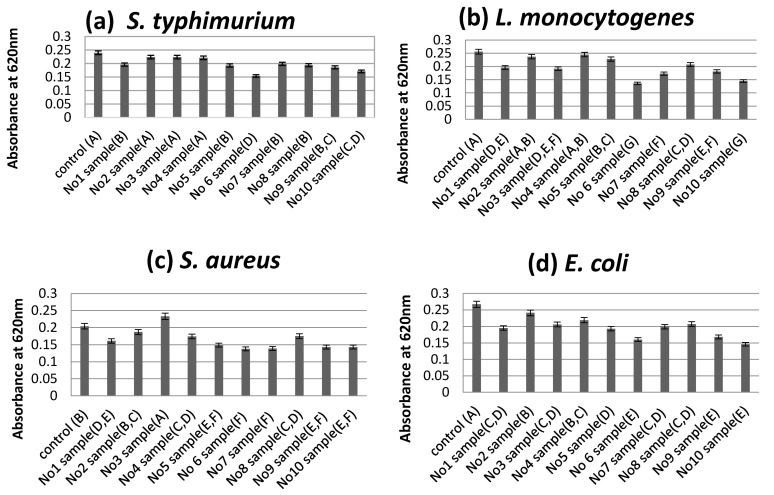
Optical density of the growth medium of *S. typhimurium* (**a**), *L. monocytogenes* (**b**), *S. aureus* (**c**), *E. coli* (**d**), *C. perfringens* (**e***), L. plantarum* (**f**), *L. putida* (**g**), *C. utilis* (**h**) and *C. jejuni* (**i**), measured the reduction of the absorbance at 620 nm as an effect of the addition of each sample at concentrations of 10 mg/mL, compared to a control (0 mg/mL) without antimicrobial extracts. The mixtures of extracts in each sample number described in Table 6; Sample means that do not share a letter are significantly different at *p* ≤ 0.05 estimated by Tukey Pairwise Comparison.

**Figure 3 plants-10-01757-f003:**
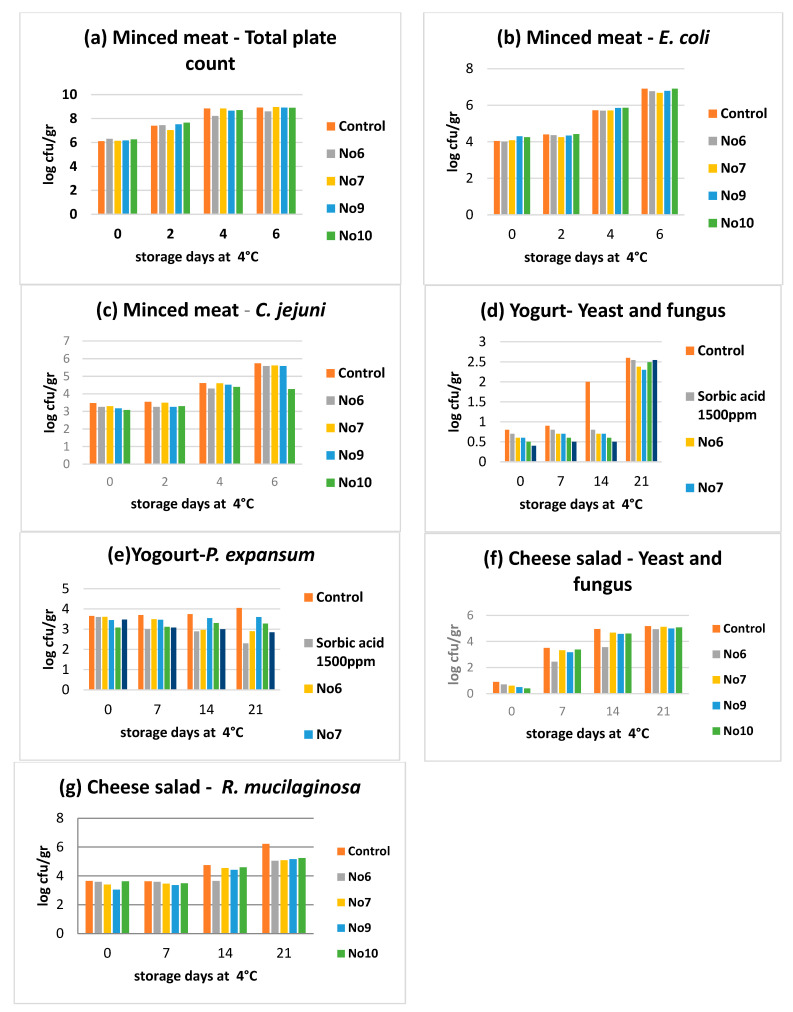
In vivo antimicrobial tests. (**a**), Minced meat—Total plate count; (**b**), Minced meat—*E. coli*; (**c**), Minced meat—*C. jejuni*; (**d**), Yogurt—Yeast and fungus; (**e**), Yogurt—*P. expansum*; (**f**), Cheese salad—Yeast and fungus; (**g**), Cheese salad—*R. mucilaginosa*.

**Figure 4 plants-10-01757-f004:**
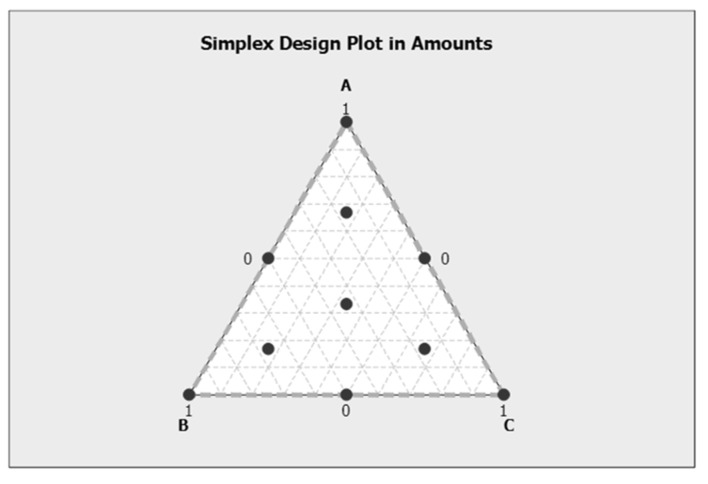
Simplex design plot in amounts.

**Table 1 plants-10-01757-t001:** Total polyphenols content (TPC) and IC_50_ of DPPH• of the examined extracts’ powders.

Number of Samples *	TPC (mgGAE/g Extract DW)	ΕC_50_ of DPPH (μg/mL)
No1	20.9 ± 1.11 ^e^	1410 ± 19 ^f^
No2	10.13 ± 1.86 ^f^	2200 ± 21 ^h^
No3	7.08 ± 1.44 ^f^	2950 ± 28 ^j^
No4	8.45 ± 1.36 ^f^	2750 ± 35 ^i^
No5	31.2 ± 1.26 ^d^	1200 ± 19 ^e^
No6	77.54 ± 4.56 ^a^	38.3 ± 2.3 ^a^
No7	43.77 ± 2.11 ^c^	650 ± 11 ^d^
No8	19.35 ± 1.47 ^e^	1520 ± 17 ^g^
No9	54.59 ± 3.14 ^b^	310 ± 14 ^b^
No10	42.57 ± 2.89 ^c^	450 ± 16 ^c^

TPC total polyphenols content, DW extract dry weight. Different letters indicate differences in the means within each column. Means that do not share a letter are significantly different in *p* ≤ 0.05. * mixtures of extracts in each sample described in Table 6.

**Table 2 plants-10-01757-t002:** Well diffusion assay of fungus, based on the of mycelium growth in mm at 3, 5 and 7 days of extract (samples No1 to No10). Mean values were estimated from four measurements.

Tested Microorganism	Days	Zone of Inhibition per Sample Number (in mm)									
		No1	No2	No3	No4	No5	No6	No7	No8	No9	No10
*A. niger*	3	5.76 ± 0.18 ^c,d^	5.77 ± 0.16 ^c,d^	6.35 ± 0.15 ^a,b^	6.18 ± 0.15 ^a,b,c^	5.81 ± 0.19 ^c,d^	4.37 ± 0.16 ^e^	5.44 ± 0.11 ^d^	6.00 ± 0.14 ^b,c^	6.53 ± 0.13 ^a^	5.49 ± 0.15 ^d^
	5	10.72 ± 0.12 ^c^	11.63 ± 0.14 ^b^	10.39 ± 0.09 ^c^	11.63 ± 0.19 ^b^	11.50 ± 0.15 ^b^	8.33 ± 0.17 ^d^	6.35 ± 0.16 ^e^	10.51 ± 0.15 ^c^	12.19 ± 0.12 ^a^	10.3 ± 0.17 ^c^
	7	12.55 ± 0.08 ^c^	14.35 ± 0.09 ^a^	12.39 ± 0.25 ^c^	14.07 ± 0.16 ^a,b^	13.65 ± 0.13 ^b^	11.82 ± 0.19 ^d^	12.41 ± 0.11 ^c^	13.62 ± 0.19 ^b^	13.8 ± 0.14 ^b^	14.14 ± 0.32 ^a,b^
*P. expansum*	3	5.53 ± 0.15 ^d,e^	6.52 ± 0.14 ^a^	5.94 ± 0.17 ^b,c,d^	6.54 ± 0.13 ^a^	6.16 ± 0.23 ^a,b,c^	4.59 ± 0.19 ^f^	5.61 ± 0.08 ^d,e^	5.91 ± 0.21 ^c,d^	5.13 ± 0.16 ^e^	6.41 ± 0.15 ^a,b^
	5	10.51 ± 0.15 ^c^	10.39 ± 0.39 ^c^	11.7 ± 0.61 ^a^	10.3 ± 0.22 ^c^	11.5 ± 0.11 ^a,b^	10.72 ± 0.16 ^b,c^	6.35 ± 0.16 ^d^	12.19 ± 0.27 ^a^	10.33 ± 0.23 ^c^	11.63 ± 0.22 ^a^
	7	13.62 ± 0.37 ^b^	12.39 ± 0.15 ^c^	13.8 ± 0.19 ^a,b^	14.14 ± 0.11 ^a,b^	13.65 ± 0.12 ^b^	12.41 ± 0.22 ^c^	9.34 ± 0.19 ^d^	14.07 ± 0.14 ^a,b^	12.55 ± 0.22 ^c^	14.35 ± 0.19 ^a^

Measured values are medians of three repetitions ± standard deviation; Means that do not share a letter in the same raw are significantly different at *p* ≤ 0.05 estimated by Tukey Pairwise Comparison.

**Table 3 plants-10-01757-t003:** Inhibition zones(mm) of bacterial pathogens, at 1 day, of extract (samples No1 to No10). Mean values were estimated from triplicate measurements.

Tested Microorganism	Zone of Inhibition per Sample Number (in mm)									
	No1	No2	No3	No4	No5	No6	No7	No8	No9	No10
*L.monocytogenes*	N.D.	N.D.	N.D.	N.D.	N.D.	2.95 ± 0.6 ^a^	N.D.	N.D.	2.75 ± 0.98 ^a^	N.D.
*C. perfringens*	3.34 ± 0.1 ^c,d,e^	3.03 ± 0.12 ^e,f^	2.75 ± 0.11 ^f^	1.70 ± 0.19 ^g^	3.65 ± 0.09 ^b,c^	3.27 ± 0.08 ^d,e^	4.6 ± 0.13 ^a^	3.67 ± 0.16 ^b,c^	3.93 ± 0.09 ^b^	3.43 ± 0.07 ^c,d^

N.D.: not detected; Measured values are medians of three repetitions ± standard deviation; Means that do not share a letter in the same raw are significantly different at *p* ≤ 0.05 estimated by Tukey Pairwise Comparison.

**Table 4 plants-10-01757-t004:** MIC/MBC assay results of avocado peel and seed extract and pomegranate peel samples against bacterial pathogens. Values are estimated as means of triplicate measurements.

Tested Micro-Organism	Sample Concentration mg/mL	Sample No1		Sample No2		Sample No3		Sample No4		Sample No5		Sample No6		Sample No7		Sample No8		Sample No9		Sample No10	
		MIC	MBC	MIC	MBC	MIC	MBC	MIC	MBC	MIC	MBC	MIC	MBC	MIC	MBC	MIC	MBC	MIC	MBC	MIC	MBC
*E. coli*	5	+		+		+		+		+		+		+		+		+		+	
	10	+		+		+		+		+		+		+		+		+		+	
	25	+		+		+		+		+		−	+	+		+		−	+	−	+
	50	+		+		+		+		+		−	+	+		+		−	+	−	+
*S. aureus*	5	+		+		+		+		+		+		+		+		+		+	
	10	+		+		+		+		+		+		+		+		+		+	
	25	+		+		+		+		+		−	+	−	+	+		−	+	−	+
	50	−	+	+		+		+		−	+	−	+	−	+	+		−	+	−	+
*S. typhimurium*	5	+		+		+		+		+		+		+		+		+		+	
	10	+		+		+		+		+		+		+		+		+		+	
	25	+		+		+		+		+		+		+		+		+		+	
	50	+		+		+		+		+		−	+	+		+		+		−	+
*L. monocytogenes*	5	+		+		+		+		+		+		+		+		+		+	
	10	+		+		+		+		+		+		+		+		+		+	
	25	+		+		+		+		+		+		+		+		+		+	
	50	+		+		+		+		+		−	+	+		+		−	+	−	+
*C. perfringens*	5	+		+		+		+		+		+		+		+		+		+	
	10	+		+		+		+		+		+		+		+		+		+	
	25	+		+		+		+		+		+		+		+		+		+	
	50	+		+		+		+		+		−	+	+		+		+		−	+
*C. jejuni*	5	+		+		+		+		+		+		+		+		+		+	
	10	+		+		+		+		+		+		+		+		+		+	
	25	−	+	−	+	−	+	−	+	−	+	−	+	−	+	−	+	−	+	−	+
	50	−	+	−	+	−	+	−	+	−	+	−	+	−	+	−	+	−	+	−	+
*L. plantarum*	5	+		+		+		+		+		+		+		+		+		+	
	10	+		+		+		+		+		+		+		+		+		+	
	25	−	+	−	+	−	+	−	+	−	+	−	+	−	+	+		−	+	−	+
	50	−	+	−	+	−	+	−	+	−	+	−	+	−	+	−	+	−	+	−	+
*A. niger*	5	+		+		+		+		+		+		+		+		+		+	
	10	+		+		+		+		+		+		+		+		+		+	
	25	−	+	+		+		+		−	+	−	+	−	+	+		−	+	+	
	50	−	+	+		+		+		−	+	−	+	−	+	−	+	−	+	+	
*P. expansum*	5	+		+		+		+		+		+		+		+		+		+	
	10	+		+		+		+		+		+		+		+		+		+	
	25	+		+		+		+		+		+		−	+	+		−	+	+	
	50	+		−	+	+		+		+		−	+	−	+	+		−	+	+	
*P. putida*	5	+		+		+		+		+		+		+		+		+		+	
	10	+		+		+		+		+		+		+		+		+		+	
	25	+		+		+		+		+		+		+		+		+		+	
	50	+		+		+		+		+		+		+		+		+		+	
*C. utilis*	5	+		+		+		+		+		+		+		+		+		+	
	10	+		+		+		+		+		+		+		+		+		+	
	25	+		+		+		+		+		+		+		+		+		+	
	50	+		+		+		+		+		−	+	−	+	+		+		−	+

(+) indicates growth of the microorganism; (−) no growth microorganism.

**Table 5 plants-10-01757-t005:** Pearson correlation of Total phenol content of samples with the MIC of the tested microorganism.

Tested Micro-Organism	Pearson Correlation with TPC	*p*-Value
*E. coli*	0.8	0.006
*S. aureus*	0.759	0.011
*S. typhimurium*	0.652	0.041
*L. monocytogenes*	0.8	0.006
*C. perfigenes*	0.652	0.041
*A. niger*	0.542	0.106
*C. utilis*	0.691	0.027

Significantly correlated in *p*-Value ≤ 0.

**Table 6 plants-10-01757-t006:** Design Table (randomized) with the concentrations, combinations.

Number of Samples	Pomegranate Peel (PP) %	Avocado Peel (AP) %	Avocado Seed (AS) %
1	16.7	66.7	16.7
2	0	100	0
3	0	0	100
4	0	50	50
5	33.3	33.3	33.3
6	100	0	0
7	50	50	0
8	16.7	16.7	66.7
9	66.7	16.7	16.7
10	50	0	50

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
