# Peer review of "The In Vitro and In Vivo Synergistic Antimicrobial Activity Assessment of Vacuum Microwave Assisted Aqueous Extracts from Pomegranate and Avocado Fruit Peels and Avocado Seeds Based on a Mixtures Design Model"

_plants, 2021, doi:10.3390/plants10091757_

Round 1

Reviewer 1 Report

Dear authors/editor

Enclosed you will find my review of the manuscript “The in vitro and in vivo Antimicrobial Activity Assessment of Vacuum Microwave-Assisted Aqueous Extracts from Pome- 3 granate and Avocado Fruits Peels and Avocado Seeds”. This work aimed to assess the potential synergist antimicrobial activity of aqueous vacuum microwave-assisted extracts from avocado peels, seeds, and pomegranate peels in liquid and powder form in vitro against foodborne bacteria, yeasts, and fungi based on a mixture design experiment.

The title of the work does not match with the objective, please rewrite the title emphasizing that you are performing mixtures of extracts and that you are expecting a synergist effect.

Introduction

The paragraph between lines 98-109 is repetitive, please rewrite this paragraph or delete it.

In line 93 it says … vacoum microwave … it should say vacuum microwave

Results

In line 112 it says … it’ is clear… it should say it is clear

In line 118 it says P-value = 0.00. This is incorrect (statistically ) it should say P-value < 0.001

All the tables and some figures (i.e. Figure 2) describe the mixtures of extracts as “number of samples”. Please change this label for Mixture of extracts* adding a footnote in the table showing that mixtures of extracts are described in Table 6

Please, perform statistical analysis for Tables 2, and 3, and for Figure 2

In Table 5 change the comma-separated numbers for the decimal points.

Discussion

As the authors pointed out, the effect of the extract mixture can be attributed to diverse compounds in the extracts. These compounds also may vary among waste samples from avocado and pomegranate peels because of differences in maturity, season, varieties. Would you expect similar results with other samples of avocado and pomegranate peels?

Author Response

COVER LETTER

Dear editor,

we would like to thank the reviewers for the efforts they made to improve this article. We made all the proposed from reviewers corrections and we are open to correct all other next possible observations, Please check our responds in the observations.

REVIEWER 1

Dear reviewer,

thank you very much for your focus observations that make us able to improve our paper.

“Enclosed you will find my review of the manuscript “The in vitro and in vivo Antimicrobial Activity Assessment of Vacuum Microwave-Assisted Aqueous Extracts from Pomegranate and Avocado Fruits Peels and Avocado Seeds”. This work aimed to assess the potential synergist antimicrobial activity of aqueous vacuum microwave-assisted extracts from avocado peels, seeds, and pomegranate peels in liquid and powder form in vitro against foodborne bacteria, yeasts, and fungi based on a mixture design experiment.”

Reviewer 1, Comments

Corrections

1.     The title of the work does not match with the objective, please rewrite the title emphasizing that you are performing mixtures of extracts and that you are expecting a synergist effect.

Dear reviewer, thank you for the observation. The title was corrected according to your observation as: “The in vitro and in vivo Synergistic Antimicrobial Activity Assessment of Vacuum Microwave Assisted Aqueous Extracts from Pomegranate and Avocado Fruits Peels and Avocado Seeds based on a mixtures design model”.

2.     The paragraph between lines 98-109 is repetitive, please rewrite this paragraph or delete it.

The paragraph was deleted as was suggested.

3.     In line 93 it says … vacoum microwave … it should say vacuum microwave

The word “vacuum” was corrected.

4.     In line 112 it says … it’ is clear… it should say it is clear

Thank you for the observation. The sentence in line 112 was corrected.

5.     In line 118 it says P-value = 0.00. This is incorrect (statistically ) it should say P-value < 0.001

The proposed correction was performed in line 118.

6.     All the tables and some figures (i.e. Figure 2) describe the mixtures of extracts as “number of samples”. Please change this label for Mixture of extracts* adding a footnote in the table showing that mixtures of extracts are described in Table 6

The addition of the proposed footnote was done in all tables and figure 2.

7.     Please, perform statistical analysis for Tables 2, and 3, and for Figure 2

ANOVA was performed and significant differences (p ≤ 0.05) were estimated by Tukey Pairwise Comparison and Tables 2, 3, and Figure 2 has been corrected.

8.     In Table 5 change the comma-separated numbers for the decimal points.

The proposed by the reviewer changes in Table 5 have been included.

9.     As the authors pointed out, the effect of the extract mixture can be attributed to diverse compounds in the extracts. These compounds also may vary among waste samples from avocado and pomegranate peels because of differences in maturity, season, varieties. Would you expect similar results with other samples of avocado and pomegranate peels?

The manuscript presents that the antimicrobial activity is attributed mainly to the polyphenols content and the occurrence of specific polyphenolic compounds.

As the reviewer observes, the content of these compounds varied based on their degree of fruits maturity and the environmental plantation conditions. On the other hand, since there is an optimized extraction methodology as described in previously published papers

Skenderidis, P.; Leontopoulos, S.; Petrotos, K.; Giavasis, I. Optimization of Vacuum Microwave-Assisted Extraction of Pomegranate Fruits Peels by  the Evaluation of Extracts’ Phenolic Content and Antioxidant Activity. Foods2020, 9, doi:10.3390/foods9111655.

Skenderidis, P.; Leontopoulos, S.; Petrotos, K.; Giavasis, I. Vacuum Microwave-Assisted Aqueous Extraction of Polyphenolic Compounds from Avocado (Persea americana) Solid Waste. Sustainability 2021, 13, 1–18, doi:10.3390/su13042166.

the extraction process is standardized and the produced powders can have the same TPC concentration but future research is needed in order answer properly to your observation about the effect of the occurrence of specific polyphenolic compounds.

In the conclusions section, we have added a sentence in order to cover the reviewer observation.

Reviewer 2 Report

The topic of manuscript is very interesting and of great importance for the further scientific researches. The manuscript is well organized, the materials and methods are described in details and provide all the information necessary for the repeatability of the experiment. Just one suggestion in methods is necessary to complete data. Information in spectroscopic methods should be supplemented with the name / type of equipment / origin / production, this applies to, for example, "Determination of Total Phenols Content (TPC) of the extracts", "Determination of Antioxidant Capacity of the samples (DPPH method). Please justify why in the experiment avocado and pomegranate peels were used, while as far as the seeds are concerned, only avocado. In addition, please indicate in the introduction part what limitations in the use of fruit seeds have in the use of food technology - please include the description of the content of anti-nutritive compounds, e.g. amygdalin in the seeds of the fruit. In terms of the description of the experiment in the food model (“Preparation of foods used in the food model”), the characteristics of the products used in the study should be completed. The nutritional value / composition of these products should be meticulously detailed - this applies to all tested models: Minced meat, cheese salad dressing and yogurt Some of the summary is too general. It should be developed with detailed conclusions that result from the research. Statistical analysis and the results obtained should be taken into account. It is necessary to clarify the conclusions and not to write generally about the properties of polyphenols from waste products.

Author Response

COVER LETTER

Dear editor,

we would like to thank the reviewers for the efforts they made to improve this article. We made all the proposed from reviewer's corrections and we are open to correct all other next possible observations, Please check our response in the observations.

REVIEWER  2

Dear reviewer,

thank you very much for your focus observations that make us able to improve our paper.

“The topic of manuscript is very interesting and of great importance for the further scientific researches. The manuscript is well organized, the materials and methods are described in details and provide all the information necessary for the repeatability of the experiment.”

Reviewer 2 Comments

Corrections

1.       Just one suggestion in methods is necessary to complete data. Information in spectroscopic methods should be supplemented with the name / type of equipment / origin / production, this applies to, for example, "Determination of Total Phenols Content (TPC) of the extracts", "Determination of Antioxidant Capacity of the samples (DPPH method).

Dear reviewer, thank you for the observation. We have added the related information in 4.5 and 4.6 sections.

2.       Please justify why in the experiment avocado and pomegranate peels were used, while as far as the seeds are concerned, only avocado.

Dear reviewer thank you very much for your observation. We would like to inform you that this paper presents only one stage of the total five sub-researches stages on the valorization of avocado and pomegranate process industry by-products. So far the results of the two sub-researches have been published

Skenderidis, P.; Leontopoulos, S.; Petrotos, K.; Giavasis, I. Optimization of Vacuum Microwave-Assisted Extraction of Pomegranate Fruits Peels by  the Evaluation of Extracts’ Phenolic Content and Antioxidant Activity. Foods2020, 9, doi:10.3390/foods9111655.

Skenderidis, P.; Leontopoulos, S.; Petrotos, K.; Giavasis, I. Vacuum Microwave-Assisted Aqueous Extraction of Polyphenolic Compounds from Avocado (Persea americana) Solid Waste. Sustainability 2021, 13, 1–18, doi:10.3390/su13042166.

And the results of the other two sub-researches are in a review process

1. “Manuscript ID: molecules-1281296”  is focused on the effects of pomegranate peels and avocado peels and seeds supplemented in corn silage on meat quality and growth rate in broiler chicken and presents a subject of which a patent is pending.

2. “Manuscript ID: IJFS-1029 presents the results from the study on Pomegranate and Avocado Residual Peel and Seed Extracts Used as Alternative and Potential Antimicrobial Compound for Suppression of Plant Pathogens.

The choice to use the avocado seeds has to do with the initial design of our research that was based on the literature review that reported that seeds have antimicrobial effects. We personally think that is fully understandable that the use of solid waste of pomegranate and avocado industry in bulk in a sustainable way for the production of natural food preservatives provide a very strong hypothesis aiming to minimize by-products wastes and contributing to sustainability. This has been reported in the last two sentences of the introduction section.

3.       In addition, please indicate in the introduction part what limitations in the use of fruit seeds have in the use of food technology.

The description of the contained anti-nutritive compounds and the reported limitations have added in the introduction section

4.       Please include the description of the content of anti-nutritive compounds, e.g. amygdalin in the seeds of the fruit.

5.       In terms of the description of the experiment in the food model (“Preparation of foods used in the food model”), the characteristics of the products used in the study should be completed. The nutritional value / composition of these products should be meticulously detailed - this applies to all tested models: Minced meat, cheese salad dressing and yogurt.

Extra information about the preparation of the foods used in the food models was added.

The nutritional value of the three foods used was added in supplementary table S1.

6.       Some of the summary is too general. It should be developed with detailed conclusions that result from the research. Statistical analysis and the results obtained should be taken into account. It is necessary to clarify the conclusions and not to write generally about the properties of polyphenols from waste products.

Dear reviewer thanks for your observation, we change the conclusions to cover your observation.

Round 2

Reviewer 1 Report

The authors have addressed all the comments and queries.